# Curcumin Mitigates Gut Dysbiosis and Enhances Gut Barrier Function to Alleviate Metabolic Dysfunction in Obese, Aged Mice

**DOI:** 10.3390/biology13120955

**Published:** 2024-11-21

**Authors:** Gopal Lamichhane, Femi Olawale, Jing Liu, Da-Yeon Lee, Su-Jeong Lee, Nathan Chaffin, Sanmi Alake, Edralin A. Lucas, Guolong Zhang, Josephine M. Egan, Yoo Kim

**Affiliations:** 1Department of Nutritional Sciences, Oklahoma State University, Stillwater, OK 74078, USA; gopal.lamichhane@okstate.edu (G.L.); femi.olawale@okstate.edu (F.O.); dayeon.lee@okstate.edu (D.-Y.L.); crystal.lee10@okstate.edu (S.-J.L.); nathan.chaffin@okstate.edu (N.C.); sanmi.alake@okstate.edu (S.A.); edralin.a.lucas@okstate.edu (E.A.L.); 2Department of Animal and Food Sciences, Oklahoma State University, Stillwater, OK 74078, USA; jing.liu12@okstate.edu (J.L.); zguolon@okstate.edu (G.Z.); 3Laboratory of Clinical Investigation, National Institute on Aging, Baltimore, MD 21224, USA; eganj@grc.nia.nih.gov

**Keywords:** curcumin, aging, gut microbiota, gut integrity, inflammation, liver health

## Abstract

This study investigates the effects of dietary curcumin on gut dysbiosis and impaired gut integrity caused by a high-fat, high-sugar diet (HFHSD) in aged male mice. Our results show that curcumin supplementation increases beneficial gut microbes and decreases harmful bacteria, leading to reduced gut inflammation and improved expression of markers of gut barrier integrity. Additionally, curcumin supports bile homeostasis in the context of aging and HFHSD consumption. These findings suggest that curcumin could be a promising dietary intervention for improving gut health in obesity and aging.

## 1. Introduction

The composition of the gut microbiome plays a crucial role in regulating both gut and metabolic health [1]. This is demonstrated by the observation that obese individuals exhibit a distinct microbiota composition, characterized by reduced diversity compared to lean individuals, with significant differences in functional potential between these groups [2]. The influence of the gut microbiome on metabolic outcomes has been further substantiated by fecal microbiota transfer studies. In these studies, germ-free mice that receive fecal microbiota from obese donors develop an obese phenotype, whereas the transfer of microbiota from healthy donors to patients with metabolic syndrome results in improved biomarkers of metabolic health [1,3]. These divergent outcomes are largely attributed to the microbiome’s effects on gut barrier function, endotoxin production, dietary fiber fermentation to produce short-chain fatty acids (SCFAs), bile acid homeostasis, and other microbial-derived metabolites that affect inflammation and energy metabolism [4,5,6]. Thus, modulating or controlling the composition of the gut microbiota could be a promising strategy to maintain gut health and mitigate metabolic diseases.

Several factors influence the composition of the gut microbiota, including perinatal microbial exposure, host genetics, immunity, antibiotic use, and diet [7]. Among these, diet is a major modifiable factor capable of inducing significant changes in the gut microbiome. Variations in gut transit time, pH, macronutrient composition, and the presence of phytochemicals in different diets can lead to substantial differences in microbial colonization [7,8,9]. For instance, mice fed a high-fat diet show an increased relative abundance of Firmicutes and a decrease in Bacteroidetes compared to control mice [10], while a high-fiber diet leads to an increase in fiber-degrading microbes such as *Bifidobacterium* and *Lactobacillus* [11]. The types of bacteria present in the gut greatly influence host health. Resistant starches, non-starch polysaccharides, and oligosaccharides that are indigestible by the host undergo microbial degradation in the gut to produce SCFAs, primarily acetate, propionate, and butyrate [7]. These metabolites support host health through various mechanisms, including promoting glucose and lipid homeostasis, modulating the immune system, protecting neurons, reducing inflammation, and offering protection against colorectal cancer, diabetes, and cardiovascular diseases [12,13]. SCFAs also improve intestinal barrier function by regulating the expression of tight junction proteins and enhancing the production of antimicrobial peptides [14].

A high-fat diet (HFD), representative of a Western-style diet, negatively impacts gut health by increasing intestinal permeability, damaging the intestinal mucosal barrier, reducing the expression of tight junction proteins, stimulating the release of hydrophobic bile acids, increasing the translocation of lipopolysaccharide (LPS) and the activation of toll-like receptor 4 (TLR4), and promoting oxidative stress in epithelial cells [15,16]. Additionally, HFD triggers proinflammatory signaling by altering cytokine release, increasing levels of tumor necrosis factor (TNF)-α, interleukin (IL)-1β, IL-6, and interferon-γ, while decreasing mRNA expression levels of anti-inflammatory cytokines such as IL-10, IL-17, and IL-22 [17]. A HFD also diminishes beneficial microbiota, such as *Lactobacillus*, *Bifidobacterium*, *Bacteroidetes*, and *Akkermansia* species, while increasing the abundance of harmful microbes like *Desulfovibrio*, which are associated with reduced barrier integrity [15]. The microbial changes induced by a high-fat, high-sugar diet (HFHSD) can also elevate the risk of gastrointestinal cancer [18].

In addition to a Western-style diet, aging significantly impacts the composition and diversity of the gut microbiota, and these changes are increasingly recognized as potential indicators of biological aging [19,20]. This prompted López-Otín and colleagues to revise the previously proposed nine hallmarks of aging to twelve in 2023, incorporating dysbiosis as one of the new hallmarks [21]. Aging leads to alterations in the gut microbiota, including decreased microbial diversity, an increased Firmicutes to Bacteroides ratio, and a decline in *Bifidobacteria*, accompanied by an increased abundance of subdominant bacteria [22]. In young adults, the gut is colonized by a diverse array of commensal microbes. However, with aging, both the population and diversity of commensal microbes decrease, often due to an increase in pro-inflammatory microbes [23]. The microbiome profile in aging individuals is less resilient and more susceptible to alterations by external factors such as diet, medication, and lifestyle, leading to long-lasting changes, whereas, in young adults, the impact of these external factors on the gut microbiome is minimal and transient [24]. Besides that, chronic low-grade inflammation under aging (inflammaging) makes the elderly population more susceptible to the negative effect of dysbiosis [24,25,26]. Aging also slows metabolic processes, affecting gut motility and nutrient absorption, which creates an environment that fosters dysbiosis and exacerbates metabolic disturbances [27]. As a result, the effects of gut dysbiosis in metabolic disorders are more severe and persistent in older individuals compared to younger ones [28]. This results in diminished epithelial cell integrity, potentially leading to leakage of microbes and endotoxins into the bloodstream, triggering systemic inflammation and predisposing individuals to age-associated diseases [23].

Supplementing the diet with bioactive food compounds has the potential to counteract these detrimental changes in microbiome composition [29]. Compounds such as anthocyanins, hesperidin, naringin, berberine, allicin, baicalein, catechins, ellagitannins, betacyanins, lycopene, kaempferol, resveratrol, and other polyphenols and alkaloids have been shown ameliorative effects on gut dysbiosis [29,30,31]. Curcumin, a polyphenol derived from turmeric, has also demonstrated beneficial effects in managing gut dysbiosis. In 2021, Li et al. reported that curcumin supplementation increased SCFA levels and the abundance of beneficial bacteria while reducing endotoxin-producing *Desulfovibrio* bacteria and serum LPS in six-week-old mice [32]. Several other studies have also highlighted curcumin’s role in alleviating dysbiosis in various mouse models [6,33,34,35,36,37]. Despite these findings, research is limited regarding whether curcumin can effectively mitigate the cumulative effects of aging and HFHSD on the gut microbiome. As our previous research demonstrated a beneficial effect of curcumin in aging-associated metabolic disease, we were curious whether it could help mitigate age-associated dysbiosis (one of the hallmarks of aging) under metabolic stress [38,39,40,41]. Therefore, this research aims to evaluate the protective effects of curcumin on gut barrier function and microbial composition in an aged mouse model subjected to an HFHSD.

## 2. Materials and Methods

### 2.1. Animals and Treatment

All animal experiments were approved by the Animal Care and Use Committee (ACUC) of the National Institute on Aging (NIA) and the Institutional Animal Care and Use Committee (IACUC) at Oklahoma State University. Eighteen- to twenty-one-month-old aged male C57BL/6 mice were obtained from the NIA-Aged Rodent Colony and housed at Charles River Laboratories (Frederick, MD, USA, or Raleigh, NC, USA). The mice were acclimatized for a week at either the NIA intramural housing facility (Baltimore, MD, USA) or the Animal Care Facilities at Oklahoma State University (Stillwater, OK, USA), with *ad libitum* access to a standard chow diet and water. Fecal samples were collected from each mouse before and after the completion of the 15-week intervention. Mice were divided into four groups (*n* = 9–10 per group): normal chow diet (NCD), curcumin-supplemented (4 g/kg) normal chow diet (NCD+CUR), HFHSD, and curcumin-supplemented (4 g/kg) HFHSD (HFHSD+CUR) groups, based on baseline body weight and 6-h fasting blood glucose levels. This dose of curcumin is equivalent to a 2 g/day dose of curcumin for a 60 kg adult based on an equivalent surface area dosage conversion method and had been used safely in mice in a previous study [39]. Details of diet composition are provided in Appendix A. *Ad libitum* access to a customized diet (purchased from Dyets Inc., Bethlehem, PA, USA) and water was provided throughout 8-week (for phenotype, Insulin tolerance test (ITT), food intake, gut integrity, and hepatic qPCR study) and 15-week (for gut microbiome profile) study periods, with weekly monitoring of food intake and body weight. The duration of treatment was decided based on previous studies [6,41].

### 2.2. Insulin Tolerance Test

Insulin tolerance test (ITT) was performed on mice after an 8-week intervention, following the methods described by Lee et al. [38]. Briefly, blood glucose levels were measured in mice that had been fasted for 6 h, at 0, 15, 30, 60, 90, and 120 min after a 0.75 IU/kg body weight insulin injection (Novo Nordisk Inc., Plainsboro, NJ, USA). Blood glucose level and area under the curve (AUC) were calculated to determine any differences in insulin sensitivity between groups.

### 2.3. Microbial Analysis of the Feces

Genomic DNA (gDNA) was isolated from fecal samples collected before and after the 15-week dietary intervention and stored at −80 °C using the QIAamp Fast DNA Stool Mini Kit (Qiagen, Hilden, Germany). The isolated gDNA was sent to DNA Link (Los Angeles, CA, USA) for 16S rRNA sequencing. The sequencing data were deposited in the National Center for Biotechnology Information (NCBI) Sequence Read Archive (SRA) under the BioProject accession number PRJNA1165253. Analysis was performed as described by Lamichhane et al. (2024) [34]. Briefly, paired sequencing reads were analyzed in QIIME 2 (v. 2020. 11). Adaptor, barcode, and primer sequences were removed using the Cutadapt plugin, followed by joining forward and reverse reads and performing quality control. High-quality reads were then denoised using the Deblur algorithm (v. 2022.8.0) to generate Amplicon Sequence Variants (ASVs). ASVs were classified using the Ribosomal Database Project (RDP) 16S rRNA training set (v. 18) and the Bayesian classifier [42]. A bootstrap confidence of 80% was used to classify taxa, and ASVs below this threshold were labeled as “_unidentified” at the highest confidently assigned taxonomic level. Any ASVs appearing in <5% of the sample were excluded from downstream analysis. Linear discriminant analysis (LDA) effect size (LEfSe) with an all-against-all multiclass analysis (*p* < 0.05) and a logarithmic threshold of 3.0 was used to determine the differential enrichment of bacterial features between groups.

### 2.4. Real-Time Quantitative Polymerase Chain Reaction (qPCR)

Total RNA was extracted from the ileum, colon, and liver tissues (*n* = 6/group) using TRIzol reagent (Thermo Fisher Scientific, Pleasanton, CA, USA), following the procedures described by Lamichhane et al., 2024 [41]. RNA quality was checked using a Nanodrop spectrophotometer and agarose gel electrophoresis. The extracted RNA was reverse-transcribed using the iScript™ cDNA Synthesis Kit (Bio-Rad Laboratories Inc., Hercules, CA, USA). The relative abundance of genes encoding for pro-inflammatory markers (TNF-α, Il-1β, and Il-6), anti-inflammatory markers (Il-10) and hepatic/biliary homeostasis-related markers (FGFR4, β-Klotho, FXRα, and BSEP) was assessed by quantitative real-time polymerase chain reaction (qRT-PCR) using SYBR Green chemistry on a CFX Opus 384 Real-Time PCR System (Bio-Rad Laboratories). The forward and reverse primer sequences used are listed in Appendix A. Relative mRNA abundance was calculated using the 2^−ΔΔCt^ method, with glyceraldehyde-3-phosphate dehydrogenase (GAPDH) or 18S as the invariant control.

### 2.5. Immunoblotting Analysis

Proteins were extracted from the ileum tissue homogenates (*n* = 3/group) using radioimmunoprecipitation assay (RIPA) buffer containing 0.5% protease and phosphatase inhibitors. Protein concentrations were determined using the bicinchoninic acid (BCA) assay, and 20 µg of protein was loaded for sodium dodecyl sulfate-polyacrylamide gel electrophoresis (SDS-PAGE). Gels were transferred onto polyvinylidene fluoride (PVDF) membranes, and transfer accuracy was confirmed by Ponceau staining. Membranes were blocked with 5% non-fat milk and incubated overnight with primary antibodies (occludin, claudin-1, β-actin; Thermo Fisher, Waltham, MA, USA). The blots were then washed with phosphate-buffered saline (PBS), incubated with a horseradish peroxidase (HRP)-linked secondary antibody (Cell Signaling Technology, Danvers, MA, USA), and washed again. Blots were visualized using Pierce enhanced chemiluminescence (ECL) Western Blotting Substrate (Thermo Fisher). Images were captured with a FluorChem R Imaging System (ProteinSimple, San Jose, CA, USA), and band intensity was quantified using ImageJ software, v 1.8.0 (National Institute of Health, Rockville, MD, USA) and normalized to β-actin.

### 2.6. Histological Analysis

Formalin-fixed jejunum and colon tissues were dehydrated in an ethanol gradient (70% ethanol, 80% ethanol, 95% ethanol and 100% ethanol) and toluene using an automated tissue processor (Shandon Citadel 2000, Waltham, MA, USA). The tissues were embedded in paraffin blocks, and 5-μm sections were cut using a microtome (Leica Biosystems, Wetzlar, Germany) and transferred to charged slides. The slides were stained with hematoxylin and eosin (H&E), and structural changes in the villi and intestinal crypts were accessed using a microscope at 10× magnification. Photomicrographs were acquired using BZ-X800 software (Keyence, Osaka, Japan).

### 2.7. Statistical Analysis

All data were analyzed using GraphPad Prism (V 9.5.1: GraphPad Inc., San Diego, CA, USA). A two-way repeated-measured analysis of variance (ANOVA) was used for body weight, food intake, and ITT. An unpaired t-test was used to analyze the area under the curve (AUC) of ITT, qPCR, and immunoblotting results. All data are presented as mean ± standard error of the mean (SEM), and statistical significance was determined at *p* ≤ 0.05. An outlier test was performed with α = 0.05 to remove any outliers

## 3. Results

### 3.1. Curcumin Supplementation Reduces Body Weight and Improves Insulin Sensitivity in Aged Mice

We observed a reduction in body weight gain in aged mice fed an HFHSD+CUR from the onset of the intervention, which persisted throughout the 8-week study (Figure 1A). The difference in mean body weight gain on weeks 7 and 8 was 3.1 g and 2.5 g, respectively. The greatest difference in mean body weight gain occurred in week 5, with a mean difference of 4.2 g. We decided to terminate the study in the 8th week as the mice began showing saturation in body weight under HFHSD feeding. We did not observe a significant difference in food intake or food efficiency ratio within dietary groups, but a negative food efficiency ratio was evident in the NCD+CUR group (Figure 1B). A slight reduction in body weight gain was also noted in the NCD+CUR group compared to the NCD group, with a mean difference of 1.5 g by week 8. Additionally, curcumin supplementation improved insulin sensitivity in both the NCD and HFHSD groups (Figure 1C).

### 3.2. Curcumin Supplementation Alters Beta Diversity of the Microbiome in Aged Mice

A marked reduction in the observed ASVs was noticed in both HFHSD and HFHSD+CUR-fed mice at the end of the 15-week dietary intervention compared to ASV levels before treatment, indicating a detrimental effect of HFHSD on microbial diversity (Figure 2A). Curcumin supplementation did not rescue this decline in ASVs, possibly due to intrasubject variation. Neither Pielou’s evenness index nor the Shannon index showed a significant change in alpha diversity after the 15-week treatment period, possibly due to variability in individual baseline microbiome, which diluted the treatment effect by masking a subtle change in alpha diversity (Figure 2B,C). However, curcumin supplementation caused significant alterations in beta diversity, reflecting a notable shift in microbial composition. Pairwise comparisons of weighted and unweighted UniFrac distances using fecal samples collected before and after treatment revealed significant differences in beta diversity across all dietary groups (Figure 2D,E).

### 3.3. Curcumin Supplementation Modifies Microbiota Composition in Aged Mice

At the phylum level, curcumin supplementation led to an increase in *Proteobacteria* and *Verrucomicrobiota* in both NCD and HFHSD groups while reducing the abundance of *Desulfobacteria* (Figure 3A,B). Furthermore, slight enrichment in the abundance of *Bacteroidota* was observed under HFHSD consumption, but curcumin supplementation mitigated this increase. Curcumin consumption had no notable effect on the enrichment of *Firmicutes* and *Bacteroidetes*.

At the genus level, the relative abundance of *Parabacteroides*, *Mucispirillum*, *Muribaculum*, *Occillospiraceae*, *Christensenellaceae*, and *Lachnospiraceae* was higher in the HFHSD+CUR group compared to HFHSD mice, while the abundance of *Alistipes*, *Muribaculaceae*, and *Bacteriodes* was lower (Figure 3C). In the NCD+CUR group, the relative abundance of *Parabacteroides*, *Dancaniella*, *Lactobacillus*, and *Christensenellaceae* was higher compared to the NCD group. Conversely, the abundance of *Alistipes*, *Muribaculaceae*, *Bacteriodes*, and *Chlostridium* was lower in the NCD-fed mice. Analysis based on ASVs revealed that curcumin-supplemented mice had a higher abundance of *Parabacteroides*, *Mucispirillum*, *Christensenellaceae*, and *Akkermansia* in both dietary conditions (Figure 3D).

LEfSe analysis revealed a higher abundance of beneficial microbiota, such as *Parabacteriodes*, *Flintibacter*, *Oscillibacter*, *Oscillospiraceae*, and *Lachnospiraceae* in the HFHSD+CUR group compared to HFHSD. In contrast, the abundance of *Muribaculaceae*, *Bilophila*, *Odoribacter*, and *Duncanjella* was higher in the HFHSD group (Figure 4B). In the NCD+CUR group, the abundance of *Parabacteriodes*, *Lachnispiraceae*, *Flintibacter*, and *Muribaculaceae* was higher compared to NCD, whereas the abundance of *Alistipes*, *Christensenellaceae*, *Clostridium*, and *Clostridales* was greater in the NCD group (Figure 4A).

### 3.4. Curcumin Supplementation Preserves Gut Architecture, Reduces Inflammation, and Enhances Tight Junction Protein Expression

Histological analysis of jejunum revealed villus atrophy under HFHSD feeding, which was ameliorated by curcumin supplementation (Figure 5A). The result was further supported by qPCR results, where curcumin supplementation significantly decreased IL-1β and showed a decreasing trend in TNF-α mRNA level (Figure 5B) while increasing the trend in anti-inflammatory IL10 expression level. Furthermore, curcumin significantly increased the expression of the tight junction protein occludin (OCLN) in the ileum of HFHSD+CUR-fed mice (Figure 5C).

However, the effect of curcumin on the colon was not as distinct as in the small intestine, indicated by no visible changes in the colon across groups (Figure 6A). Also, qPCR analysis of inflammatory markers showed no significant difference in pro-inflammatory or anti-inflammatory gene expression in the colon of HFHSD-fed mice supplemented with curcumin (Figure 6B). However, significant downregulation of proinflammatory TNF-α was observed in curcumin-supplemented NCD-fed mice (Figure 6B).

### 3.5. Curcumin Supplementation Ameliorates Bile Acid Homeostasis-Related Markers in the Liver

There was a notable increase in the expression of farnesoid X receptor α (FXRα) and bile salt export pump (BSEP) in the NCD+CUR group compared to the NCD group (Figure 7A). Curcumin supplementation also significantly increased the expression of β-Klotho and FXRα gene in the livers of HFHSD-fed mice (Figure 7B). Additionally, curcumin led to an upregulating trend in the expression of fibroblast growth factor receptor 4 (FGFR4) and BSEP genes in the livers of HFHSD+CUR-fed mice.

## 4. Discussion

The Western diet (WD), rich in fat and sucrose, is associated with gut dysbiosis—a disruption in the balance of gut microbiota [43]. This dysbiosis contributes to increased intestinal permeability (“leaky gut”) and systemic inflammation, which can negatively affect various tissues, including the liver [44]. Since the liver directly receives blood from the digestive tract through the portal vein, it is particularly susceptible to the consequences of a leaky gut, such as liver inflammation and metabolic dysfunction-associated steatotic liver disease (MASLD) [45]. This issue is exacerbated in older individuals, as aging has been shown to worsen gut dysbiosis, promoting a cycle of inflammation and declining health, as previously reported by our groups [46,47]. In this study, we aim to investigate how curcumin supplementation mitigates liver damage in aged mice fed an HFHSD by modulating the gut-liver axis.

Our results demonstrate that curcumin supplementation effectively mitigated gut dysbiosis, attenuated weight gain, reduced gut inflammation, and enhanced gut integrity in aged obese mice. These improvements in gut health were accompanied by enhanced liver metabolic function, particularly with regard to the regulation of bile acid homeostasis. Notably, the reduction in body weight was biologically significant, as evidenced by a marked improvement in insulin sensitivity in the HFHSD+CUR group compared to the HFHSD group, as observed during the insulin tolerance test (ITT). The alterations in microbial populations further underscore the profound impact of diet and diet-induced metabolic changes, including body weight, on the gut microbiota. Curcumin increased the abundance of Bacteroidetes in obese aged mice, a microbial shift associated with the production of short-chain fatty acids (SCFAs) such as acetate, propionate, and butyrate. Propionate, in particular, plays a key role in reducing fat accumulation by inhibiting hepatic lipogenesis and promoting satiety [48]. These findings support earlier studies showing that curcumin can improve body weight and composition in obesity, likely through its effects on gut microbiota [49,50].

Both aging and WD consumption contribute to gut dysbiosis, but their combined effects have not been thoroughly characterized. In our study, HFHSD further aggravated gut dysbiosis in aged mice, marked by a lower relative abundance of beneficial bacterium such as *Parabacteroides*, *Occillospiraceae*, *Mucispirillum*, *Muribaculum*, *Flintibacter*, *Lachnospiraceae*, and *Akkermansia*, and a higher abundance of *Desulfobacteria*. The latter is associated with inflammatory conditions, including inflammatory bowel disease (IBD) [51]. The beneficial bacteria that declined are known for producing SCFAs, such as butyrate, which support colon health, provide energy for colonocytes, and exert anti-inflammatory effects [14]. The loss of these bacteria may increase intestinal permeability, allowing toxins and bacteria to enter the bloodstream [52]. Curcumin selectively modulates certain microbial populations, influencing the composition of microbial communities without significantly affecting overall species richness or evenness. Notably, the curcumin-supplemented group showed a lower abundance of *Desulfobacteria* and a higher abundance of beneficial bacteria like *Parabacteroides*, *Mucispirillum*, and *Flintibacter*, suggesting a positive shift in the gut microbiota toward a healthier composition.

In this study, HFHSD-fed aged mice exhibited increased levels of pro-inflammatory cytokines (such as TNF-α and IL-1β) in the ileum, which were reduced following curcumin supplementation. Previous studies have also demonstrated a correlation between *Desulfobacteria* and elevated secretion of inflammatory factors [53,54]. These bacteria release metabolites and LPS that activate immune cells, leading to the release of pro-inflammatory cytokines. Our results support the idea that curcumin-induced shifts in the gut microbiota play a significant role in reducing chronic low-grade gut inflammation. We observed a more pronounced effect in the ileum compared to the colon, likely due to the higher density of immune cells, including Peyer’s patches, in the ileum, making it more responsive to treatment [55]. Additionally, curcumin has low systemic absorption, and its concentration may be higher in the ileum than the colon due to differences in transit time and degradation, leading to a stronger anti-inflammatory effect in the ileum [33].

Additionally, curcumin supplementation increased the expression of the tight junction protein occludin in Ileum, which had been reduced in HFHSD-fed aged mice. This finding is consistent with previous studies by Tian et al. [56]. They demonstrated curcumin’s protective role in intestinal ischemia-reperfusion injury through modulation of zonula occludens-1 (ZO-1) protein expression and downregulation of the TNF-α pathway. In our study, curcumin increased gut integrity in the ileum. Overall, these results suggest that curcumin supports gut integrity in aged obese mice by preserving tight junctions and reducing gut inflammation.

A compromised gut barrier allows inflammatory mediators to enter the bloodstream, where they can travel to the liver and trigger an inflammatory response [57]. In our study, curcumin restored the expression of key liver markers, such as FXRα and β-Klotho, which had been downregulated in the obese aged mice. These markers are linked to bile acid homeostasis and overall liver function. The increased levels of FXRα and β-Klotho in curcumin-supplemented groups may be attributed to the improved gut microbiota, as *Parabacteroides*—which increased following curcumin supplementation—has been shown to alleviate obesity-related dysfunctions and activate intestinal gluconeogenesis and FXR signaling by generating succinate and secondary bile acids. The FXR signaling pathway is crucial for bile acid homeostasis, and curcumin’s ability to modulate this pathway may explain its protective effects on liver health. Yang et al. [58] previously proposed that curcumin exerts its effects against cholestasis by restoring bile acid homeostasis and reducing inflammation through an FXR-dependent mechanism. Our findings support this hypothesis, demonstrating that curcumin improves liver function and reduces inflammation in aged obese mice by modulating gut dysbiosis, reducing chronic low-grade gut inflammation, enhancing gut barrier integrity, and preserving bile acid homeostasis via FXR signaling.

One limitation of this study was the relatively short duration of treatment, which may explain the lack of significant changes in insulin sensitivity despite observable trends. Also, the use of the same cohort for gut microbiome analysis and gut integrity study could provide a more reliable correlation between microbiome change and improved gut health. Analysis of circulating inflammatory markers could provide strong evidence on whether curcumin-mediated reduced low-grade inflammation in the gut was strong enough to manifest in the systemic circulation. Future studies should explore the long-term effects of curcumin supplementation in aged obese mice.

## 5. Conclusions

In conclusion, we demonstrated that curcumin supplementation effectively mitigates the negative effects of an HFHSD on gut health in aged obese mice. Curcumin reduced weight gain, improved gut microbiota composition by increasing beneficial bacteria and reducing harmful bacteria, and enhanced gut integrity by promoting the expression of tight junction proteins. These effects were accompanied by a significant reduction in ileum inflammation, as evidenced by decreased expression levels of pro-inflammatory cytokines. Furthermore, curcumin modulated genes regulating bile acid homeostasis in the liver, likely through the FXR signaling pathway, which may play a key role in improving liver metabolic function. Overall, curcumin supplementation holds promise as a dietary intervention to protect against gut dysbiosis, inflammation, and disrupted bile acid homeostasis associated with diet-induced obesity in aging.

## Figures and Tables

**Figure 1 biology-13-00955-f001:**
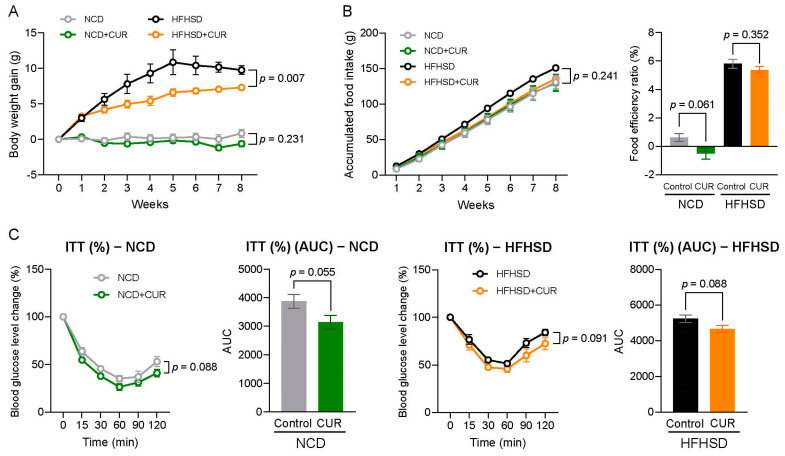
Curcumin supplementation mitigates body weight gain and improves insulin sensitivity in aged male mice. (**A**) Body weight gain (g), (**B**) accumulated food and food efficiency ratio, and (**C**) insulin tolerance test. (*n* = 9 for NCD and NCD+CUR; *n* = 6 for HFHSD; *n* = 9 for HFHSD+CUR). Results are expressed as mean ± SEM.

**Figure 2 biology-13-00955-f002:**
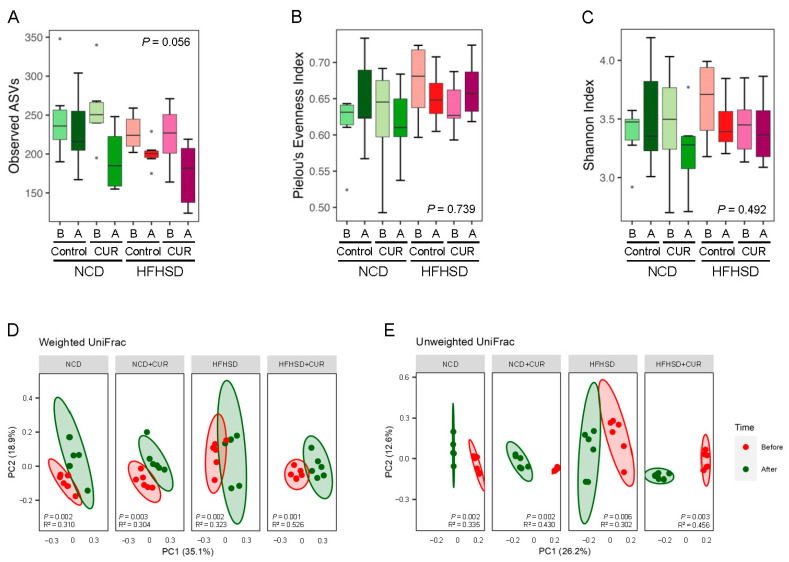
Curcumin supplementation alters the beta diversity of the gut microbiome in the feces of aged male mice. (**A**) Observed ASVs, (**B**) Pielou’s evenness index, (**C**) Shannon index, (**D**) weighted UniFrac, and (**E**) unweighted UniFrac analysis of the feces from mice. (*n* = 6 per group). For (**A**–**C**): B, baseline and A, after the 15-week intervention. Dots above and below the bar graph in (**A**–**C**) represents outliers.

**Figure 3 biology-13-00955-f003:**
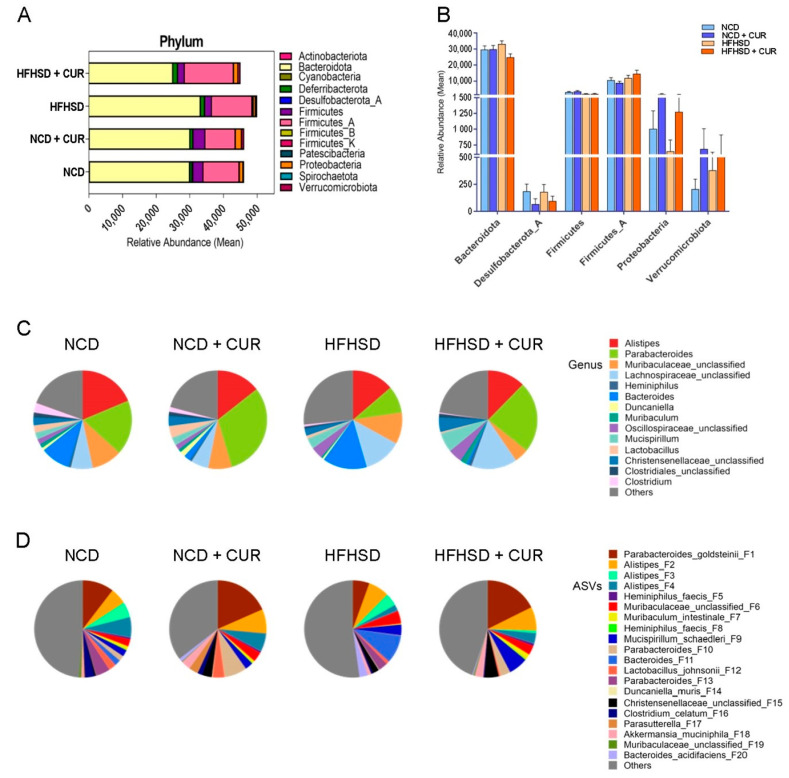
Curcumin supplementation alters the relative abundance of the gut microbiome in the feces of aged male mice. The figure shows the relative abundance of the microbiome at (**A**,**B**) phylum level, (**C**) genus level, and (**D**) ASV level (*n* = 6 per group).

**Figure 4 biology-13-00955-f004:**
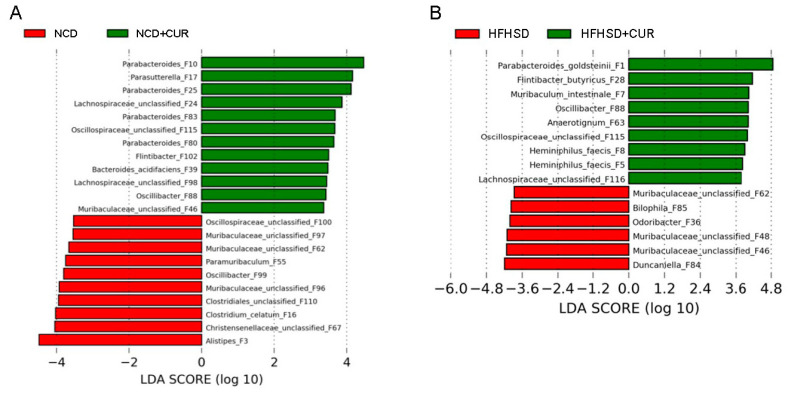
LEfSe analysis of the abundance of the gut microbiome in the feces of aged male mice. The figure shows the relative abundance of the microbiome in (**A**) NCD vs. NCD+CUR and (**B**) HFHSD vs. HFHSD+CUR (*n* = 6 per group).

**Figure 5 biology-13-00955-f005:**
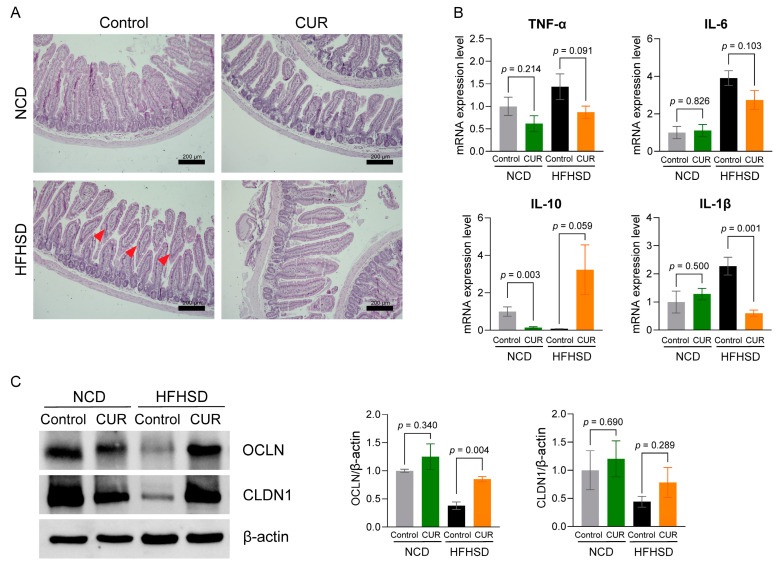
Curcumin improves the integrity of the small intestine by decreasing inflammation and increasing the expression of tight junction proteins in aged mice. (**A**) Representative images from H&E staining of jejunum tissues (scale bar = 200 µm), (**B**) expression of inflammatory markers in the ileum, and (**C**) expression of tight junction proteins in the ileum of mice. The results are presented as mean ± SEM, *n* = 6/group. Red arrow in the histological section highlights villous atrophy, which was not observed in HFHSD+CUR group.

**Figure 6 biology-13-00955-f006:**
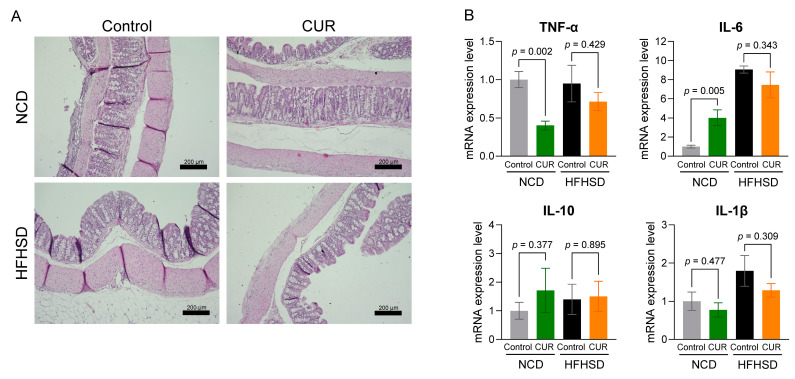
Curcumin has a modest effect on the colon of aged mice. (**A**) Representative images from H&E staining of the colon (scale bar = 200 µm), (**B**) expression of inflammatory markers in the colon of aged mice. The results are presented as mean ± SEM, *n* = 6/group.

**Figure 7 biology-13-00955-f007:**
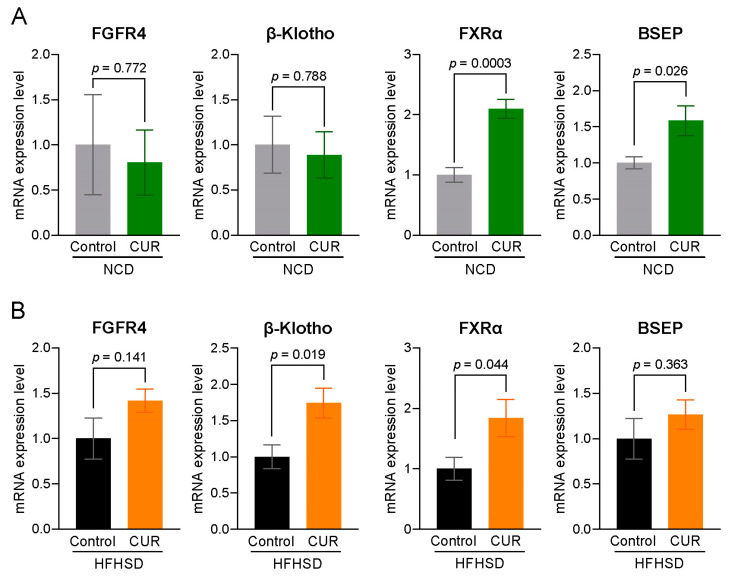
Curcumin supplementation regulates biliary homeostasis-related genes in aged male mice. (**A**) Relative expression in NCD vs. NCD+CUR and (**B**) relative expression in HFHSD vs. HFHSD+CUR. Results are expressed as mean ± SEM, *n* = 6 per group.

## Data Availability

The raw sequencing reads for 16S rRNA sequencing were deposited in the NCBI Sequence Read Archive database under the same BioProject accession number, PRJNA1165253. All the datasets used and/or analyzed during the current study are available from the corresponding authors upon reasonable request.

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
