# Peer review of "Curcumin Mitigates Gut Dysbiosis and Enhances Gut Barrier Function to Alleviate Metabolic Dysfunction in Obese, Aged Mice"

_biology, 2024, doi:10.3390/biology13120955_

Round 1

Reviewer 1 Report

Comments and Suggestions for Authors

The authors have tried to understand the influence of curcumin on obesity and aging using animal model. The article is written well. The figures are well described. 

Overall the work is interesting. However, the following queries have to be addressed before making a decision on the publication of this article

1. The choice of treatment method - 4g/Kg body weight of curcumin - how was this arrived at, the toxic level is to be included in the study
2. The authors have concentrated only on gut and its function as the title suggests, but to be a holistic research paper, information on the toxicity of curcumin on kidney tissues is to be included, when liver marker are studied, kidney markers to be included as well. 

3. The survival of the mice is to be provided

Though the article is of interest, these information should be provided before considering for publication 

Author Response

We appreciate the reviewer for recognizing the significance of our study and providing us with helpful comments. In the revised manuscript, we have diligently addressed all the points raised, which are highlighted in yellow.

Reviewer 1

Comment 1: The choice of treatment method - 4g/Kg body weight of curcumin - how was this arrived at, the toxic level is to be included in the study.

Author response: Thank you for your valuable comment. The 4 g/kg (0.4% w/w) dose of curcumin is equivalent to a 2 g/day for a 60 kg adult, calculated using the surface area dosage conversion method. This dose has been safely used in previous studies without reporting toxic effects. We have now clarified this in the manuscript (section 2.1, lines 146-149). Additionally, the safety of this dose has been supported by earlier research, which did not indicate any adverse effects at this level.

Comments 2: The authors have concentrated only on gut and its function as the title suggests, but to be a holistic research paper, information on the toxicity of curcumin on kidney tissues is to be included, when liver marker are studied, kidney markers to be included as well.

Author response: Thank you so much for your comment. We acknowledge the importance of kidney function in overall health. However, this study specifically focuses on gut health and the potential involvement of the gut-liver axis as a key determinant of gut health. Thus, kidney health markers were not included in the scope of this study. Importantly, the dose of curcumin used in this study has been shown to be safe in previous research, and we did not anticipate or observe kidney toxicity. Supporting this, there was no significant difference in kidney weight among the study groups (data not shown in the manuscript). While kidney health was outside the primary scope of this study, we agree that investigating the impact of curcumin on kidney function would provide valuable insights. We plan to explore this aspect in future studies.

Comments 3: The survival of the mice is to be provided

Author response: Thank you for highlighting the importance of survival data, particularly in aging-related research. However, this study was a short-term investigation and did not include a longitudinal survival analysis. Nonetheless, we confirm that all the mice survived until the end of the study period, with the exception of one mouse in the NCD control group, which was euthanized due to severe dermatitis.

Reviewer 2 Report

Comments and Suggestions for Authors

Overall, the study presents an interesting investigation into the potential protective effects of curcumin on gut health in the context of aging and an obesogenic diet. However, several questions for further exploration remain.

1. Why was curcumin chosen over other polyphenols or dietary supplements known to influence the gut microbiome?

2. Could the authors provide more context on how aging exacerbates gut dysbiosis in metabolic disorders and how this effect differs in younger populations?

3. Could the authors justify the 15-week duration as sufficient to observe the long-term benefits of curcumin in the context of aging and metabolic dysfunction?

4. What is the biological relevance of the reported weight differences in mice? Could this difference be attributed to factors other than curcumin supplementation?

5. Can the authors explain the biological implications of significant changes in beta diversity but not alpha diversity? Does this indicate that curcumin affects the relative abundance of specific bacteria rather than overall diversity?

6. in result 2, Pielou's Evenness and Shannon Index do not show significant changes after treatment. The authors attribute this to high intersubject variability. However, it would be beneficial to explain why curcumin did not influence alpha diversity as much as expected, especially given the significant shifts in beta diversity. Adding more details on how intersubject variability may have impacted the results could strengthen this point.

7. Why did curcumin have a stronger anti-inflammatory effect in the ileum compared to the colon? Could this indicate a site-specific effect of curcumin on gut inflammation?

8. Could the authors expand on the proposed mechanism by which curcumin modulates FXR signaling? Are there other signaling pathways that might be involved?

9. There is limited discussion on how curcumin’s effects might vary based on different doses or delivery methods. How did the author determine the 0.4% as the intervention dose?

10. Why did curcumin affect the gut microbiota? The underlying mechanism?

Author Response

We appreciate the reviewer for recognizing the significance of our study and providing us with helpful comments. In the revised manuscript, we have diligently addressed all the points raised, which are highlighted in yellow.

Reviewer 2

Comment 1: Why was curcumin chosen over other polyphenols or dietary supplements known to influence the gut microbiome?

Author response: Thank you for your insightful comment. Curcumin, a widely studied polyphenol derived from turmeric, was chosen because of its well-documented ability to influence the gut microbiome in various contexts, including studies in young mice (Li et al., 2021). Additionally, curcumin is known for its anti-inflammatory and antioxidant properties, which are particularly relevant in mitigating age-associated conditions. Our research group has extensively studied curcumin's effects on aging-associated diseases, including metabolic dysfunctions and inflammatory conditions (Lamichhane et al., 2024; Lee et al., 2023; Lee et al., 2022). Building on this foundation, we aimed to explore whether curcumin could ameliorate gut dysbiosis—one of the hallmarks of aging—under conditions of metabolic stress in aged mice. We have included this rationale in the last paragraph of the introduction for added clarity and context (lines 126-129).

References

Li, S.; You, J.; Wang, Z.; Liu, Y.; Wang, B.; Du, M.; Zou, T. Curcumin Alleviates High-Fat Diet-Induced Hepatic Steatosis and Obesity in Association with Modulation of Gut Microbiota in Mice. Food Research International 2021, 143, 110270, doi:10.1016/j.foodres.2021.110270.

Lamichhane, Gopal, et al. "Curcumin-Rich Diet Mitigates Non-Alcoholic Fatty Liver Disease (NAFLD) by Attenuating Fat Accumulation and Improving Insulin Sensitivity in Aged Female Mice under Nutritional Stress." Biology 13.7 (2024): 472.

Lamichhane G, Liu J, Lee SJ, Lee DY, Zhang G, Kim Y. Curcumin mitigates the high-fat high-sugar diet-induced impairment of spatial memory, hepatic metabolism, and the alteration of the gut microbiome in Alzheimer’s disease-induced (3xTg-AD) mice. Nutrients. 2024 Jan 12;16(2):240.

Lee DY, Lee SJ, Chandrasekaran P, Lamichhane G, O’Connell JF, Egan JM, Kim Y. Dietary Curcumin Attenuates Hepatic Cellular Senescence by Suppressing the MAPK/NF-κB Signaling Pathway in Aged Mice. Antioxidants. 2023 May 27;12(6):1165.

Lee SJ, Chandrasekran P, Mazucanti CH, O’Connell JF, Egan JM, Kim Y. Dietary curcumin restores insulin homeostasis in diet-induced obese aged mice. Aging (Albany NY). 2022 Jan 1;14(1):225.

Comment 2: Could the authors provide more context on how aging exacerbates gut dysbiosis in metabolic disorders and how this effect differs in younger populations?

Author response: Thank you for your comment. Aging significantly alters the microbiome profile, making it less resilient and more prone to long-lasting changes in response to external factors such as diet, medication, and lifestyle. In younger populations, similar external factors typically induce only transient and minimal effects on the gut microbiome.

Furthermore, aging is associated with chronic low-grade inflammation, known as inflammaging, which exacerbates the negative impacts of gut dysbiosis and metabolic disorders. This state of heightened susceptibility makes the elderly population more vulnerable to gut microbiome disruptions compared to younger individuals. We have included this explanation in the manuscript for clarity (lines 102–108).

Comment 3: Could the authors justify the 15-week duration as sufficient to observe the long-term benefits of curcumin in the context of aging and metabolic dysfunction?

Author response: Thank you for raising this point. In this study, we observed significant outcomes, including reduced weight gain and enhanced insulin sensitivity, within the 15-week period. These findings align with our previous studies using the same dietary regimen, which demonstrated improvements in MASLD, insulin signaling, and hepatocellular senescence over the same duration (Kim et al., 2019; Lee et al., 2023; Lee et al., 2022).

While this duration was sufficient to capture key metabolic improvements, we acknowledge that longer-term studies could provide additional insights into curcumin’s effects on aging and metabolic dysfunction. We have now clarified this in the manuscript (lines 126–129).

Reference

  1. Kim Y, Rouse M, González-Mariscal I, Egan JM, O’Connell JF. Dietary curcumin enhances insulin clearance in diet-induced obese mice via regulation of hepatic PI3K-AKT axis and IDE, and preservation of islet integrity. Nutrition & metabolism. 2019 Dec;16:1-1.
  2. Lee DY, Lee SJ, Chandrasekaran P, Lamichhane G, O’Connell JF, Egan JM, Kim Y. Dietary Curcumin Attenuates Hepatic Cellular Senescence by Suppressing the MAPK/NF-κB Signaling Pathway in Aged Mice. Antioxidants. 2023 May 27;12(6):1165.
  3. Lee SJ, Chandrasekran P, Mazucanti CH, O’Connell JF, Egan JM, Kim Y. Dietary curcumin restores insulin homeostasis in diet-induced obese aged mice. Aging (Albany NY). 2022 Jan 1;14(1):225.

Comment 4: What is the biological relevance of the reported weight differences in mice? Could this difference be attributed to factors other than curcumin supplementation?

Author response: Thank you for this important question. The observed weight differences are biologically relevant, as they were accompanied by improved insulin sensitivity in the HFHSD+CUR group compared to the HFHSD group, as demonstrated in the insulin tolerance test (ITT). These findings suggest that curcumin supplementation mitigated the adverse metabolic effects of the HFHSD.

The significant microbial shifts observed in our study further support the role of curcumin-induced metabolic changes on the gut microbiota. Baseline measurements were taken, and mice were randomly assigned to groups to minimize confounding factors. Additionally, all animals were housed under identical controlled conditions throughout the study, ensuring the observed differences were primarily attributable to curcumin supplementation. We have expanded this discussion in the manuscript (lines 343–357).

Comment 5: Can the authors explain the biological implications of significant changes in beta diversity but not alpha diversity? Does this indicate that curcumin affects the relative abundance of specific bacteria rather than overall diversity?

Author response: The significant changes in beta diversity but not alpha diversity suggest that curcumin selectively modulates the composition of microbial communities without altering overall species richness or evenness. Alpha diversity metrics (e.g., Shannon Index and Pielou’s Evenness) measure the number and balance of species within a sample, while beta diversity captures differences in community composition between groups.

Our results indicate that curcumin influences specific bacterial taxa, such as promoting beneficial bacteria (e.g., Parabacteroides and Flintibacter) and suppressing pathogenic ones (e.g., Alistipes and Clostridales), leading to shifts in community structure reflected in beta diversity. This targeted modulation underscores curcumin’s impact on microbial ecology rather than broad-spectrum alterations in diversity. This explanation has been added to the discussion (lines 252–254, 343–357).

Comment 6: in result 2, Pielou's Evenness and Shannon Index do not show significant changes after treatment. The authors attribute this to high intrasubject variability. However, it would be beneficial to explain why curcumin did not influence alpha diversity as much as expected, especially given the significant shifts in beta diversity. Adding more details on how intrasubject variability may have impacted the results could strengthen this point.

Author response: Thank you for this valuable suggestion. While curcumin significantly altered beta diversity, the absence of alpha diversity changes likely reflects the compositional rather than quantitative nature of curcumin’s effects. High intrasubject variability in baseline microbiomes likely diluted any subtle treatment-induced changes, further masking potential shifts in alpha diversity metrics.

This variability underscores the importance of individual microbiome profiles in influencing treatment outcomes. We have expanded on this point in the manuscript (lines 252–254).

Comment 7: Why did curcumin have a stronger anti-inflammatory effect in the ileum compared to the colon? Could this indicate a site-specific effect of curcumin on gut inflammation?

Author response: Thank you for this insightful question. The pronounced effects in the ileum compared to the colon may result from the higher density of immune cells and specialized immune structures, such as Peyer’s patches, in the ileum. These structures likely render the ileum more responsive to anti-inflammatory interventions like curcumin.

Additionally, curcumin’s low systemic absorption may lead to higher localized concentrations in the ileum compared to the colon due to differences in transit time and degradation. These factors may contribute to the stronger anti-inflammatory effects observed in the ileum. This has been clarified in the discussion (lines 379–384, 386).

Reference

Mörbe UM, Jørgensen PB, Fenton TM, von Burg N, Riis LB, Spencer J, Agace WW. Human gut-associated lymphoid tissues (GALT); diversity, structure, and function. Mucosal immunology. 2021 Jul 1;14(4):793-802.

Comment 8: Could the authors expand on the proposed mechanism by which curcumin modulates FXR signaling? Are there other signaling pathways that might be involved?

Author response: We focused on FXR signaling to explore the gut-liver axis, as FXR plays a crucial role in bile acid homeostasis and metabolic regulation. However, other pathways, such as the reduction of inflammation and oxidative stress through NF-κB inhibition, may also contribute to curcumin’s effects on the gut microbiota and metabolic health. This additional information has been included in the manuscript (lines 389–392).

Comment 9: There is limited discussion on how curcumin’s effects might vary based on different doses or delivery methods. How did the author determine the 0.4% as the intervention dose?

Author response: The 4 g/kg (0.4% w/w) dose of curcumin was selected based on its equivalence to a 2 g/day dose for a 60 kg adult, calculated using the surface area dosage conversion method. This dose has been widely used in previous studies without reporting toxic effects. We have clarified this in the manuscript (lines 146–149).

Comment 10: Why did curcumin affect the gut microbiota? The underlying mechanism?

Author response: Curcumin’s low systemic bioavailability means that a substantial amount remains in the gut, where it can directly influence the gut microenvironment. By modulating inflammatory pathways (e.g., NF-κB) and altering bacterial metabolic activities, curcumin reshapes the gut microbiota. For instance, curcumin may suppress pathogenic bacteria and promote beneficial ones, as supported by previous studies (Zam et al., 2018).

Reference

Zam W. Gut microbiota as a prospective therapeutic target for curcumin: A review of mutual influence. Journal of nutrition and metabolism. 2018;2018(1):1367984.

Reviewer 3 Report

Comments and Suggestions for Authors

Dear authors. Because this experiment has been carried outon mice, please indicate it in the title.

Dealing with the materials and methods please add the diet compositions it will help and for example it will help to know which sugar has been used. Fructose ?

Regarding the results presentation, please revised your redaction by indicating all the p value in the text to help the reader with the significance or the trend of the presented results. P values that must be also integrated in all the figures and not only in which ones. Please clearly indocate what is non-significant, statisticaly significant and only a trend.

at 3.4 please present separately the result by part of tissues, strains, cytokin or tight junctions i guess it will help the reader to embrace all the results. And it could be considered that strains, cytokine are tissues dependent ?

related to that the figure 5 could be separated in two to three separate figures.

Figure 6 and dedicated results. Could it be interesting to compare the significance of the two diets for all the markers, for example BSEP NCD with BSEP HFHSD ?

In the disussion, on line 339 to 352, could you please be more specific of the considered part of the gut ? And in relation with the presented results of course. On line 354 to 358 it is possible to integrate some of the results of the experiment ?

On line 375 to 380 I partially disagree, because you indicate that analysis of circulating inflammatory markers could provide strong evidence on whether low grade inflammation in the gut caused systemic inflammation. In my opinion it has alredy been demonstrated with IBS for example.

And improved bioavailable curcumin already exists and have been the subject of clinical trials.

Based on all the previous comment, I think that the conclusion could be improved to reflect the value of your results.

Author Response

We appreciate the reviewer for recognizing the significance of our study and providing us with helpful comments. In the revised manuscript, we have diligently addressed all the points raised, which are highlighted in yellow.

Reviewer 3

Comment 1: Dear authors. Because this experiment has been carried out on mice, please indicate it in the title.

Author response: Thank you for your valuable feedback. We have revised the title to explicitly indicate that the study was conducted on mice. The new title is: “Curcumin Mitigates Gut Dysbiosis and Enhances Gut Barrier Function to Alleviate Metabolic Dysfunction in Obese Aged Mice” (lines 3–4).

Comment 2: Dealing with the materials and methods please add the diet compositions it will help and for example it will help to know which sugar has been used. Fructose?

Author response: We appreciate your suggestion. We have now included the detailed diet composition in the supplementary materials (Table 2 and Table 3), specifying the types of sugars used, including fructose. This information has also been referenced in the manuscript for clarity (lines 148–149).

Comment 3: Regarding the results presentation, please revised your redaction by indicating all the p value in the text to help the reader with the significance or the trend of the presented results. P values that must be also integrated in all the figures and not only in which ones. Please clearly indicate what is non-significant, statistically significant and only a trend

Author response:  Thank you for this important observation. In response to your comment, we have updated the manuscript to include p-values for all relevant data in both the text and figures. Each figure now clearly specifies statistical significance, trends, and non-significant results. We believe these additions enhance the clarity and accessibility of the results for the readers.

Comment 4: at 3.4 please present separately the result by part of tissues, strains, cytokine or tight junctions I guess it will help the reader to embrace all the results. And it could be considered that strains, cytokine are tissues dependent?

Author response: Thank you for this insightful suggestion. To improve clarity, we have now organized the results in section 3.4 (lines 297-303 and lines 310-315) by tissue type. Specifically, we have separated the results into two categories: small intestine (jejunum and ileum) and colon. This reorganization highlights the tissue-dependent effects of strains, cytokines, and tight junctions, making the results easier for readers to follow.

Comment 5: related to that the figure 5 could be separated in two to three separate figures.

Author response: We appreciate this recommendation. Figure 5 has now been divided into two separate figures: one for the small intestine (jejunum and ileum) and another for the colon. This separation improves the visualization of tissue-specific results and aids in comprehending the differences between regions.

Comment 6: Figure 6 and dedicated results. Could it be interesting to compare the significance of the two diets for all the markers, for example BSEP NCD with BSEP HFHSD?

Author response: Thank you for pointing this out. As our study design included control comparisons within each dietary group, we have maintained this approach to preserve statistical clarity and reduce confounding. Comparing markers across diets (e.g., BSEP NCD vs. BSEP HFHSD) might introduce complexities unrelated to the primary aim of the study. We have clarified this reasoning in the revised manuscript.

Comment 7: In the discussion, on line 339 to 352, could you please be more specific of the considered part of the gut? And in relation with the presented results of course. On line 354 to 358 it is possible to integrate some of the results of the experiment?

Author response: Thank you for this detailed comment. We have revised the discussion to specify the gut regions under consideration (e.g., small intestine vs. colon) in relation to the results. Additionally, we have integrated key experimental findings (e.g., differential cytokine expression and microbial changes) into the discussion to provide a more cohesive interpretation (lines 343–357).

Comment 8: On line 375 to 380 I partially disagree, because you indicate that analysis of circulating inflammatory markers could provide strong evidence on whether low grade inflammation in the gut caused systemic inflammation. In my opinion it has alredy been demonstrated with IBS for example.

Author response: We appreciate this perspective. It is indeed well-documented that gut inflammation can lead to systemic inflammation, as demonstrated in IBS and IBD models. However, in our study, we aimed to highlight the limitation of not measuring circulating inflammatory markers. We have rephrased the sentence to: “Analysis of circulating inflammatory markers could provide strong evidence on whether curcumin-mediated reductions in low-grade gut inflammation were sufficient to manifest in systemic circulation.” This clarification reflects the limitation of our study while acknowledging existing evidence (lines 414–416).

Comment 9: And improved bioavailable curcumin already exists and have been the subject of clinical trials.

Author response: Thank you for bringing this to our attention. We agree that formulations with enhanced bioavailability, such as nanoencapsulation and emulsions, are available and have undergone clinical trials. In light of this, we have removed this limitation from the discussion.

Comment 10: Based on all the previous comment, I think that the conclusion could be improved to reflect the value of your results.

Author response: Thank you for this suggestion. We have revised the conclusion to better reflect the significance of our findings. Specifically, we emphasize curcumin’s potential in modulating gut microbiota, enhancing gut barrier function, and alleviating metabolic dysfunction in obese aged mice. The updated conclusion integrates key results and highlights the broader implications of our study.

Round 2

Reviewer 2 Report

Comments and Suggestions for Authors

The authors have addressed all my concerns with clarity, and I don't have any further comments.

Reviewer 3 Report

Comments and Suggestions for Authors

Dear authors,

Thanks for the modfications introduced.